# Translating a Thin-Film Rehydration Method to Microfluidics for the Preparation of a SARS-CoV-2 DNA Vaccine: When Manufacturing Method Matters

**DOI:** 10.3390/pharmaceutics14071427

**Published:** 2022-07-07

**Authors:** Allegra Peletta, Eakachai Prompetchara, Kittipan Tharakhet, Papatsara Kaewpang, Supranee Buranapraditkun, Nongnaphat Yostrerat, Suwimon Manopwisedjaroen, Arunee Thitithanyanont, Jonathan Avaro, Leonard Krupnik, Antonia Neels, Kiat Ruxrungtham, Chutitorn Ketloy, Gerrit Borchard

**Affiliations:** 1Section of Pharmaceutical Sciences, Institute of Pharmaceutical Sciences of Western Switzerland (ISPSO), University of Geneva, 1211 Geneva, Switzerland; allegra.peletta@unige.ch; 2Department of Laboratory Medicine, Faculty of Medicine, Chulalongkorn University, Bangkok 10330, Thailand; eakachai.p@chula.ac.th (E.P.); kankittipan12@hotmail.com (K.T.); 3Center of Excellence in Vaccine Research and Development (Chula VRC), Faculty of Medicine, Chulalongkorn University, Bangkok 10330, Thailand; kaewpangpapatsara@gmail.com (P.K.); bsuprane2001@yahoo.com (S.B.); nongnaphat.yos@gmail.com (N.Y.); rkiatchula@gmail.com (K.R.); 4Thai Pediatric Gastroenterology, Hepatology and Immunology (TPGHAI) Research Unit, Faculty of Medicine, Chulalongkorn University, Bangkok 10330, Thailand; 5Department of Microbiology, Faculty of Science, Mahidol University, Bangkok 10400, Thailand; swiboonut@gmail.com (S.M.); arunee.thi@mahidol.edu (A.T.); 6Empa, Swiss Federal Laboratories for Materials Science and Technology, Center for X-ray Analytics, Lerchenfeldstrasse 5, 9014 St. Gallen, Switzerland; jonathan.avaro@empa.ch (J.A.); leonard.krupnik@empa.ch (L.K.); antonia.neels@empa.ch (A.N.); 7Department of Chemistry, University of Fribourg, Chemin du Musée 9, 1700 Fribourg, Switzerland; 8Empa, Swiss Federal Laboratories for Materials Science and Technology, Materials-Biology Interactions Lab, Lerchenfeldstrasse 5, 9014 St. Gallen, Switzerland

**Keywords:** DNA vaccines, delivery systems, liposomes, microfluidics, thin-film layer rehydration, manufacturing method, immunogenicity, structural differences

## Abstract

Previous investigations conducted on a liposomal formulation for a SARS-CoV-2 DNA vaccine manufactured using the thin-film layer rehydration method showed promising immunogenicity results in mice. The adaptation of the liposomal formulation to a scalable and reproducible method of manufacture is necessary to continue the investigation of this vaccine candidate. Microfluidics manufacture shows high potential in method translation. The physicochemical characterization of the blank liposomes produced by thin-film layer rehydration or microfluidics were shown to be comparable. However, a difference in lipid nanostructure in the bilayer resulted in a significant difference in the hydration of the thin-film liposomes, ultimately altering their complexation behavior. A study on the complexation of liposomes with the DNA vaccine at various N/P ratios showed different sizes and Zeta-potential values between the two formulations. This difference in the complexation behavior resulted in distinct immunogenicity profiles in mice. The thin-film layer rehydration-manufactured liposomes induced a significantly higher response compared to the microfluidics-manufactured samples. The nanostructural analysis of the two samples revealed the critical importance of understanding the differences between the two formulations that resulted in the different immunogenicity in mice.

## 1. Introduction

Over the past two years, vaccine research and development significantly accelerated due to the COVID-19 pandemic. New vaccine technologies such as mRNA and viral-vectored DNA vaccines have been approved in record time. Nucleic acid vaccine technology was proven to be effective, easily tunable, and fast to produce. However, its efficacy and stability are connected to the use of suitable adjuvants and delivery systems. Despite the approval of various COVID-19 vaccines to this day, we are still far from the end of the pandemic threat [1], with one of the main challenges being vaccine accessibility and distribution in low-/middle-income countries [2]. In fact, vaccine dose hoarding by developed countries and difficult distribution and storage conditions have resulted in a mere 17.61% of the population in low-income countries having received a single dose to date (18 May 2022) [3]. The recent real-world data on vaccine effectiveness indicated that SARS-CoV-2-related symptom severity as well as transmission rates are reduced when the vaccine coverage is increased (over 60%) [4]. More importantly, the mortality rate is significantly decreased in individuals who received the booster dose [5]. Moreover, the risk–benefit assessment of the recently licensed vaccine Comirnaty clearly demonstrated that the benefits outweigh the potential risks in 16–29-year-old vaccinees [6]. Ideally, a pandemic vaccine should be affordable and easy to produce, have a high manufacturing capacity and remain stable at temperature variations [7]. Together with vaccine approval, adjuvant research has also greatly improved, with the approval of lipid-nanoparticles and Iscomatrix such as Matrix M^®^. One of the biggest challenges in the development of these delivery systems is reproducibility, scalability and the lack of precise guidelines from regulatory authorities for their manufacture [8]. Therefore, the production method is very important in order to allow for a sufficient production capacity as well as increased accessibility. The gold standard method for liposome manufacture is the thin-film rehydration method, where a lipid film is formed by the evaporation of an organic solvent and subsequently rehydrated with a specific buffer, to form homogeneous vesicles [9]. In order to obtain monodisperse samples, the liposomal suspension needs to undergo a high mechanical stress method, such as sonication [10]. The drawback of these methods is the lack of reproducibility, scalability and automation, all essential attributes when it comes to industrial production. Additional research needs to be carried out to find methods suited to meet the needs described above.

In the past years, microfluidics emerged as the method of choice for liposome manufacture [11,12]. The mixing of an organic and an aqueous phase and the resulting change of solubility conditions induce the dissolved lipids to precipitate and form vesicles. The characteristics of those vesicles can be tuned by modulating the flow rate, flow rate ratio and size of the capillaries. In a microfluidics device, the streams flow in a pre-manufactured channel under non-turbulent process conditions, allowing reproducibility, automation and size tuning. This avoids the use of harsh processes such as sonication to reduce particle size and implement the homogeneity of the sample, permitting the encapsulation of fragile molecules such as nucleic acids [13]. Previously, a cationic liposomal formulation manufactured using thin-film rehydration and complexed with a SARS-CoV-2 spike (S) protein expressing plasmid DNA (pCMVkan-S) showed promising immunogenicity results in mice [14]. In lieu of upscale and transfer, a thin-film (TF) rehydration protocol was translated to a microfluidics (MF) method.

## 2. Materials and Methods

For all experiments, the lipids 1,2-dioleoyloxy-3-trimethylammoniumpropane (DOTAP) chloride (Mw: 698.54 g/mol) and 1.2 dioleoyl-sn,glycero-3-phosphoethanolamine (DOPE) were obtained from Sigma-Aldrich (Buchs, Switzerland), whereas dipalmitoylphosphatidylcholine (DPPC) was purchased from Avanti Polar Lipids (Alabaster, AL, USA). Methanol, ethanol and isopropanol were acquired from Fisher Scientific (Leicester, UK) and Chloroform Emprove^®^ from Merck (Darmstadt, Germany). Phosphate buffer saline (PBS) Mg-/Ca- (Gibco^®^, Thermo Fisher, Grand Island, NY, USA) was used for all experiments. The AI334 Anti-Spike protein antibody was purchased from the antibody facility of the University of Geneva (Geneva, Switzerland). An FITC-labeled secondary rabbit anti-mouse antibody was obtained from Dako (Glostrup, Denmark).

### 2.1. Cell Lines

Complete-modified Eagle’s medium (C-MEM) acquired from Gibco, Thermo Fisher, was used to grow human embryonic kidney 293 cells (HEK 293, ATTC CRL-1573, LGC Standards, UK) for transfection studies. The MEM was completed with 10% fetal calf serum (FCS), 1% penicillin–streptomycin (Pen/Strep), sodium pyruvate 1mM and 0.1% non-essential amino acids (from Gibco, Thermo Fisher) to obtain C-MEM. Complete-Dulbecco’s modified Eagle’s medium (C-DMEM) acquired from Gibco, Thermo Fisher, was employed to culture the murine macrophage RAW 264.7 cell line (ATTC TIB-71) used for mitochondrial activity studies. The DMEM was completed with 10% fetal calf serum (FCS), 1% penicillin–streptomycin (Pen/Strep), sodium pyruvate 1 mM and 0.1% non-essential amino acids (from Gibco, Thermo Fisher) to obtain C-DMEM. The DMEM was also used to culture African green monkey kidney epithelial cells (Vero-E6, ATTC CRL-1586) supplemented with 2% FBS and 1% Pen/Strep. All cells were incubated at 37 °C and in an atmosphere containing 5% CO_2_.

### 2.2. Plasmid DNA Construct

Plasmid DNA construction was detailed in our previous report [15]. In brief, the gene-encoding cytoplasmic-deleted SARS-CoV-2 spike protein was optimized for human codon usage and synthesized by GenScript (Piscataway, NJ, USA). The gene was then subcloned into pCMVkan expression vector (provided by Professor Barbara K. Felber, NCI, NIH, USA), referred to as pCMVkan-S, and transformed into the *E. coli* DH5-alpha strain (Invitrogen, Carlsbad, CA, USA) for plasmid propagation. Plasmid purification was carried out with a Qiagen endotoxin-free giga plasmid kit (Hilden, Germany) following the manufacturer’s protocol. The plasmid was characterized by nucleotide sequencing and gel electrophoresis.

### 2.3. Mice Experiments

Female ICR mice at 4–5 weeks of age were procured from the National Laboratory Animals Center, Mahidol University (Thailand). After 1 week of acclimatization, the mice were randomly divided into eight groups (seven mice/group), and then immunized intramuscularly (IM) three times at two-week intervals with 100 µg pCMVkan-S complexed with liposome at different N/P ratios including 0.25:1, 1:1 and 3.2:1 prepared by thin-film rehydration (TF) or microfluidics (MF) methods. Intramuscular (IM) injection followed by electroporation (EP), IM-EP (Ichor Medical System, San Diego, CA, USA) and naked-IM pCMVkan-S injection were used as comparators. Blood samples were collected every 2 weeks after each immunization for antibody titer measurement. At 4 weeks after the last immunization, the mice were euthanized by 30% CO_2_ inhalation, and splenocytes were collected for T-cell response analysis.

### 2.4. Liposomal Formulation and pCMVkan-S Complexation

Liposomes were formulated using the TF method described elsewhere [9,14]. Briefly, lipids including DPPC, DOPE and DOTAP were dissolved in 4 mL chloroform:methanol (9:1) at a ratio of DPPC:DOPE:DOTAP (8:4:4) in a 50 mL round-bottom flask. The solvent was evaporated using a Rotavapor (Büchi, V-855, Essen, Germany) for 1 h at 190 mbar and successively rehydrated with 2 mL of warm PBS. To homogenize the sample, the suspension was subjected to 12 min of probe sonication. For manufacture by MF method, the lipids (of DPPC:DOPE:DOTAP at a molar ratio of 8:4:4) were dissolved in ethanol. Manufacture was performed by injecting the organic phase and aqueous buffer (PBS) into separate chamber inlets of the NanoAssemblr^®^ Ignite (Precision Nanosystems VA, Vancouver, BC, Canada). The flow rate ratio (FRR) (ratio between solvent and aqueous stream) varied from 2:1 to 3:1, and the total flow rate (TFR) from 12 mL/min to 4 mL/min. The ethanol was eliminated after two cycles of tangential flow filtration (TFF) performed with a Hollow Fiber Filter Module (Repligen-C02-E100-05-S) column against PBS. For the physicochemical characterization, Nano ZS Zeta-Sizer (Malvern, UK) was used to measure the size and size distribution using dynamic light scattering (DLS) and Zeta-potential. Lipoplexes were formed by mixing a suitable volume of liposome suspension calculated based on the various positively charged nitrogen to negatively charged phosphate molar ratios (N/P ratios), from 0.25:1 to 100:1, diluted in PBS. The plasmid was added into the liposome suspension dropwise. The suspension was formulated by trituration and allowed to rest for 30 min at room temperature. The particle size and Zeta-potential values were further examined. Liposome quantification was analyzed by the U-HPLC method as previously described [14].

### 2.5. Transmission Electron Microscopy (TEM)

An EMS GlowQube instrument was used to glow-discharge carbon-coated grids (Quantifoil, Hatfield, PA, USA). The grids were blotted with the liposomes sample at a concentration of 0.05 mg/mL for 1 min, and then stained twice with 50 µL of 2% uranyl acetate and finally air dried. A Talos L120C microscope (Thermo Fisher, Waltham, MA, USA) was used to acquire images and later analyzed with Fiji software (Image J, 1.53c plus, National Institutes of Health, Bethesda, MD, USA).

### 2.6. Small Angle X-ray Scattering (SAXS)

Transmission SAXS of the lipid bilayers was obtained on a Bruker NanoStar (Bruker AXS GmbH, Karlsruhe, Germany) within the Center for X-ray Analytics at EMPA St. Gallen (Switzerland). The instrument was equipped with a pinhole collimation system permitting a beam size at a sample position of about 400 μm in diameter. X-ray generation was assisted with a micro-focused X-ray Cu source (wavelength Cu K α = 1.5406 ˚A), and scattering patterns were recorded on a 2D MikroGap technology-based detector (VÅNTEC-2000 2D with 2048 × 2048 pixels and 68 × 68 μm each pixel size) along with a custom-built semi-transparent beam stop. The sample-to-detector distance was set at 27 cm and further calibrated with a silver behenate powder standard; the resolved scattering vector modulus q covers a range between 0.26 and 8.2 nm^−1^. In the WAXS configuration, the sample-to-detector distance was set at 5 cm and further calibrated with corundum powder standard; the resolved scattering vector modulus q covers a range between 4.5 and 27 nm^−1^. The scattering patterns were recorded at room temperature under moderate vacuum conditions (10^−2^ mbar) to limit air scattering. The scattering of each sample was recorded over an integration time of two times 1 h at a different position along the glass capillary to limit beam damages.

The two lipid bilayer solutions were put into glass capillaries of 1.5 mm thickness, sealed with wax at their extremities, and mounted on a vertical sample holder in the analytical chamber of the Nanostar alongside an empty capillary and PBS buffer for adequate background subtraction. Background subtraction was done systematically for all samples by normalizing the scattering intensity of the direct beam and subtracting the contribution from the glass capillary and PBS buffer to the normalized samples. The intensity of the semi-transparent beamstop from the direct beam scans was used for transmission normalization.

### 2.7. Mitochondrial Activity (WST-1)

The cell viability and proliferation were assessed using a WST-1 reagent that measures mitochondrial activity in living cells. RAW 264.7 cells were plated in a 96-well plate and incubated at 37 °C, 5% CO_2_ at a density of 1 × 10^5^ cells/mL in C-DMEM. After 24 h, 100 µL of cationic liposome suspension, diluted in C-DMEM, at a starting concentration of 7 mg/mL to 0.22 mg/mL (applying 1:1 dilutions), was added to the cells and incubated for 24 h. A positive control in the cytotoxicity experiment was prepared by adding water for injection (WFI) to the cells to induce cells lysis, whereas the negative control was represented by untreated cells. The WST-1 reagent (4-[3-(4-iodophenyl)-2-(4-nitrophenyl)-2H-5-tetrazolio]-1,3-benzene disulfonate) (Roche, Basel, Switzerland) diluted 1:10 in C-DMEM was added to the cells after medium aspiration, and then incubated for 30 min at 37 °C and 5% CO_2_. Lastly, the UV absorbance of the samples was quantified at 450 nm on a plate reader (Biotek, Synergy Mx, Waldbronn, Germania).

### 2.8. Liposomal Complex Stability

Liposome-plasmid complex stability was analyzed by 1% agarose gel electrophoresis. Twenty microliters of lipoplex were mixed with DNA loading dye^®^ and then loaded in the gel. Naked pCMVkan-S that served as a control and 1 kbp DNA ladder (Thermo Fisher, GeneRuler^®^) were run in parallel. The gel was run using Tris–borate–EDTA (TBE) buffer with a voltage of 80 V for 30 min. UV transillumination (Bio-Rad, ChemiDoc^®^ XRS, Hemel Hempstead, UK) was used to visualize the plasmid migration.

### 2.9. Transfection Studies and Immunofluorescence

The cell transfection protocol was previously described [14]. Briefly, HEK293 adherent cells, in C-MEM, cultured on glass coverslips and transfected with pCMVkan-S/liposome complexes (used to detect the specific viral protein) were incubated for 24 h. The positive control consisted of complexes of Lipofectamine 2000 (Invitrogen, CA, USA) and pCMVkan-S prepared following the manufacturer’s protocol. Negative control was prepared with pCMVkan-S alone and non-transfected cells. Transfected HEK293 were fixed with PBS + 4% formaldehyde (Reactolab, Servion, Switzerland) for 30 min at room temperature, blocked with PBS+ 40 mM ammonium chloride (NH_4_Cl) for 5 min. Cell permeabilization was performed with PBS + 0.1% Triton X-100 for 15 min, and protein blocking with PBS + 0.2% BSA (PBS-BSA) (AppliChem GmbH, Darmstadt, Germany) for 5 min. An anti-Spike protein antibody produced at the University of Geneva (Anti-S AI334) was used at a concentration of 5 ng/mL in PBS–BSA to detect S protein expression. The antibody was added onto the transfected cells and incubated for 1 h at 37 °C [16]. After three times washing with PBS–BSA, a FITC-tagged rabbit anti-mouse secondary antibody diluted 1:100 in PBS–BSA was added to the cells for 1 h at 37 °C. Three more washes with PBS–BSA were performed, and then the cells were counter stained with 1:2000 Hoechst dye (Invitrogen, CA, USA) in PBS for 10 min at room temperature, protected from light, and finally rewashed three times [11]. The cells were mounted on slides with Vectashield (Vectorlab, CA, USA) before visualization. Images were obtained using a Nikon A1r Spectral point scanning confocal microscope with a 40× oil immersion objective and analyzed with Fiji software (Image J, 1.53c plus).

### 2.10. Immunogenicity Measurement in Immunized Mice

The immune responses in the immunized mice were analyzed for both antibodies and T-cell responses. Spike-specific total IgG was measured using indirect ELISA technique by coating S-trimer onto the 96-well plates. The mid-point titers were calculated and expressed as the reciprocals of the dilution that showed an optical density (OD) at 50% of the maximum value subtracted from the background (BSA plus secondary antibody). The micro-neutralization test (MN) was carried out in the BSL-3 facility at the Department of Microbiology, Faculty of Science, Mahidol University, Thailand, to measure NAb titers against the wild-type live virus. The indirect ELISA and MN procedures were described in our previous report [14]. The mouse T-cell response was examined by the IFN-γ ELISPOT assay that was also previously detailed [14,15]. The spike peptide pools (25–26 peptides/pool) used in this study were 15 amino acids (aa) overlapping by 10 amino acids spanning the entire sequence of SARS-CoV-2 spike protein (*n* = 253). Spike-specific IFN-γ secreting splenocytes were reported as spot-forming cells (SFCs)/10^6^ after subtraction of the spots in negative control wells.

### 2.11. Statistical Analysis

Statistical analysis was performed using GraphPad Prism 9 software (San Diego, CA, USA). Comparisons of the data between the groups of samples made by TF vs. MF were made using unpaired Welch’s t-test. Comparisons among different groups and controls were carried out by two-way ANOVA with multiple comparisons. In vivo results comparisons of the data between the groups were done using Mann–Whitney tests. All *p*-values < 0.05 were defined as statistically significant.

## 3. Results

A liposomal formulation manufactured using a microfluidics (MF) device and complexed into a DNA vaccine was characterized for its physicochemical characteristics, toxicity and efficacy in inducing a specific immune response in an animal model. The results obtained were compared with a liposomal formulation made by thin-film layer (TF) rehydration, which was previously tested and found immunogenic in mice [14]. The goal of the study was to identify operative methods where the physical and chemical characteristics are similar, in order to improve the manufacturability of a product.

### 3.1. Microfluidics Method Optimization, Blank Liposomes

The variation of the parameters of the microfluidics protocol was investigated in order to obtain a size and Zeta-potential corresponding to the TF formulation, using the same lipid composition DPPC:DOPE:DOTAP (8:4:4) (DOTAP 4), with a DLS-measured size of 133.4 ± 19.8 nm and a Zeta-potential of +48 ± 12 mV. To reach the desired size and polydispersity index (PDI), various FRR and TFR were investigated (Figure 1). As seen in Figure 1, a high FRR (3:1) resulted in nanoparticle sizes of around 50 nm, which is almost three times lower than our expected size. The same result was observed for high TFR such as 12, 8 or 6 mL/min. The ideal conditions to obtain similar physicochemical parameters were FRR 2:1 and TFR 4 mL/min, resulting in a particle size of 154.1 ± 23.7 nm and Zeta-potential of +48 ± 4 mV. Size and morphology were observed by TEM after negative staining (Figure 2). The sizes and morphology of the two samples were comparable, with no major difference observed. However, the sizes obtained by TEM appeared to be bigger than those observed by DLS. In addition, the surface of the liposomes appeared to be more patterned. This might be the result of sample-drying TEM during the sample preparation, inducing the liposomes to flatten, leading to a slight increase in size and modification of their shape.

### 3.2. Toxicity of Microfluidics Manufactured Liposomes

After assessing the physicochemical characteristics of the liposomes obtained, their biocompatibility was assessed through a WST-1 assay. Cationic liposomes, and DOTAP in particular, have been reported for their cytotoxicity, which may represent a challenge. Therefore, liposome suspensions at various concentrations were incubated with murine macrophages for 4 h. The lack of cytotoxicity at high doses such as 7 mg/mL of the total formulation needed to be tested in order to prove biocompatibility at the maximum in vivo dose. All cells incubated with samples showed a mitochondrial activity above 80%, a sign of the absence of cytotoxicity (Figure 3). The incubation time of 4 h was chosen reflecting the residence time of nanoparticles in the body before antigen-presenting cell (APC) uptake. The concentration range was chosen to prove that the samples are non-cytotoxic at the maximum used concentration in vivo of 3.7 mg/mL. The results show comparable results for the two formulations for most of the concentrations, with significant differences in the positive control composed of water for injection (WFI). In addition, microfluidics-manufactured samples were also tested after an incubation of 24 h with the cells at the same concentrations, and showed no toxicity on RAW 264.7 cells (Appendix A).

### 3.3. Liposome/pCMVkan-S Complexation: Comparison with Thin-Film Rehydration-Manufactured Liposomes

The size, size distribution and Zeta-potential of the two samples were compared upon plasmid DNA complexation at various N/P ratios (Figure 4 and Figure 5). First, similar trends were observed in the two samples, with sizes below 1 µm for complexes of 0.25:1, 1:1 and 3.2:1 ratios. These samples contain an excess amount of pCMVkan-S, which explains the overall negative Zeta-potential observed in Figure 5. At higher N/P ratios, an aggregation phenomenon is observed via DLS measurement and confirmed by eye inspection of the sample, which resulted in an increase in size and PDI, showing more heterogeneous structures. In this section, we observed a switch in charge from negative to positive values. The highest N/P ratio of 100:1 resulted in a reduction of size and a positive charge, revealing an excess of cationic liposomes in the sample. Additionally, differences in the complexation behavior were observed at various N/P content, as shown in Table 1. In particular, complexes with N/P ratios of 0.25:1 and 1:1 MF-formulated nanoparticles had a superior size compared to the same sample formulated with the TF method. Moreover, in the MF-formulated sample 3.2:1, the size was smaller than for 1:1 and 0.25:1, but this was not the case for the TF samples, where the sizes of 3.2:1, 1:1 and 0.25:1 and their PDI were comparable. For the Zeta-potential figure (Figure 5), we observed that the charge shift from negative to positive values occurred at a 50:1 N/P ratio for the TF method particles, but at 25:1 for the MF-formulated complexes. Electrophoretic mobility studies were also carried out to assess the extent of pCMVkan-S complexation to liposomes (Figure 6). The two liposome formulations showed comparable results at the different N/P ratios tested. As the goal of this project was to compare the liposomes made with a different method based on a previous paper [14], the N/P ratios chosen for further investigations were 0.25:1 and 1:1. In addition, the 3.2:1 ratio was also included, as the complexation studies for the microfluidics sample showed preferable physicochemical properties.

### 3.4. SAXS Results

Small-angle scattering data of the liposome made with MF and TF do not present diffraction peaks typical for multi-layered lipid bilayers, suggesting that both liposomes formed in this study are single bilayered (Figure 7) [17]. While additional information on the structure of the lipid bilayer remains difficult to extract due to the signal-to-noise ratio, the analysis of the local lipid organization within the bilayers, via WAXS, presents some significant differences. It has to be noted that in both samples, no preferential molecular orientation was found for the lipids composing the bilayers.

The deconvolution of diffraction peaks in WAXS was performed by fitting Gaussian peaks to the experimental data (Figure 8). The number of peaks was established by considering the second derivative minima.

Liposomes prepared with MF present a single sharp peak at 18.65 nm^−1^ and a broad peak at lower q-regimes (15.65 nm^−1^). The deconvolution of the scattering signal in the wide-angle regime of the TF samples presents two sharp peaks at 16.06 and 20.29 nm^−1^ as well as a broad peak at lower q-regimes (12.77 nm^−1^). The established lipid–lipid chain distance was established by conversion of the peak position in inverse space to real space distances based on the formula:(1)$$D=\frac{2\pi }{q^{0}}$$
where *D* is the real space distance and *q*^0^ is the value of the peak maxima.

The I(*q*) plot in Figure 8 reporting the position of the lipid WAXS peak maxima, *q*^0^, gives information about the average spacing for molecular packing corresponding mainly to the chain spacing, and the width of the peak gives information about qualitative disorder in the packing. A summary of the fitting parameters can be found in Table 2.

### 3.5. Transfection Efficacy of Microfluidics Manufactured DOTAP 4 on HEK 293 Cells

Figure 9 shows the results of the transfection efficacy studies on HEK 293 cells of MF-manufactured samples at N/P ratios of 0.25:1, 1:1 and 3.2:1. N/P 1:1 and 3.2:1 samples showed similar S-protein production to the positive control lipofectamine, whereas the 0.25:1 ratio sample appeared to be of lower transfection capacity. No fluorescence was observed for the negative controls (cells alone and naked pCMVkan-S-transfected cells).

### 3.6. Immunogenicity in Mice

#### 3.6.1. Total S-Specific IgG

The immunized mice sera were analyzed for the kinetic response of S-specific IgG. The results revealed the gradual increase in IgG levels following each immunization, as seen in Figure 10. After the first dose, all liposome formulations and naked-IM injection showed a similarity in IgG titers, while IM-EP induced significantly higher IgG titers than the other groups. After receiving two doses of liposome formulations at N/P of 1:1 and 3.2:1 prepared by MF and TF methods, respectively, the IgG titers were significantly higher than the naked-IM group. After three doses of immunization, the liposome formulations and IM-EP immunization showed higher IgG titers (ranging from 1.5 to 4.8 folds) than naked-IM injection (GMT = 6,285). At the same N/P ratio, liposomes prepared by TF seemed to be higher than MF. Comparisons of the GMTs between MF and TF for N/P of 0.25:1, 1:1 and 3.2:1 were 12,597 vs. 15,704 (*p* = 0.4242), 17,181 vs. 30,096 (*p* = 0.0938) and 9,672 vs. 29,202 (*p* < 0.01). Interestingly, among the liposome groups, the N/P ratio of 3.2:1 of the MF method showed the lowest titer, while the 1:1 ratio of the TF liposomes showed the highest titer. At this timepoint (Week 6), the IgG titers induced by liposome formulations and IM-EP (GMT = 22,710) were not significantly different.

#### 3.6.2. Neutralizing Antibodies against Live Viruses

NAb titers against WT (Wuhan strain) virus were assessed 2 weeks after the mice received the second dose (Week 4) and third dose (Week 6). At Week 4, IM-EP immunization was the most potent strategy in the induction of NAb. All groups of liposome formulations showed similar NAb levels with IM injection. The NAb titers in all groups were further increased after the third dose was given. Consistent with the IgG results, TF induced higher titers than the MF method. Comparisons of the MN50 GMTs between MF vs. TF for N/P of 0.25:1, 1:1 and 3.2:1 were 320 vs. 640 (*p* = 0.5321), 861 vs. 1050 (*p* = 0.5804) and 320 vs. 2100 (*p* < 0.0134), respectively. The MN50 GMTs for naked-IM and IM-EP at this timepoint were 353 and 1159, respectively. Among the liposome formulation groups, MF at N/P of 3.2:1 showed the lowest MN50 titers, similar to those immunized by naked-IM injection.

#### 3.6.3. T-Cell Response

S-specific interferon gamma-secreting cells from mice splenocytes were analyzed at Week 4 after the third immunization. After stimulation with S-peptide pools, IM-EP induced the highest magnitude of response compared to those immunized by all liposome formulations and IM injection. Among the liposome groups, both manufactured using MF and TF methods, there was no significant increase of T-cell responses when compared to IM injection. Interestingly, an approximately two-fold lower magnitude of response was observed in two groups (N/P 0.25:1 and 3.2:1) of liposomes prepared using the MF method.

## 4. Discussion

As the antibody level, especially for neutralizing antibodies, is proven to be correlated with protective immunity against SARS-CoV-2 infection as well as a reduction in hospitalization, an immunization regimen that induces high antibody levels is desired [18,19]. In this study, using a SARS-CoV-2 DNA vaccine as a model antigen, we showed that most liposome formulations enhanced S-specific IgG levels. The induced IgG titers of N/P 1:1 and 3.2:1 prepared by TF were similar to those of IM-EP immunization after a two- or three-dose immunization (Figure 10). Consistent with the IgG results, the neutralizing antibody titers for N/P of 1:1 and 3.2:1 prepared using the TF method were comparable with IM-EP after three doses. In the case of the TF method, increasing the N/P ratio increased the antibody response (Figure 11).

The magnitude of the T-cell response of all liposome formulations was significantly lower than IM-EP (Figure 12). This is consistent with our previous finding that IM-EP very potently stimulates T cells [14]. The unique characteristic of EP may explain this finding. Previous reports demonstrated that EP does not only enhance DNA transfection, but also has an adjuvant-like property. EP enhanced inflammatory cell infiltration into the injection site [20]. Moreover, it also induced minute tissue injury as well as the activation of the danger signal through the pro-inflammatory pathway, resulting in T-cell migration [21]. These phenomena thereby significantly enhanced the immune response.

Although the limitation of a lower T-cell response was obtained in our study, liposome/DNA complexes might be of benefit under conditions where a T-cell response is not a major factor in preventing or controlling the infection [22]. Improvements to liposome production, such as including T-cell bias adjuvants in the formulation, might be required to enhance T-cell responses.

Interestingly, liposomes manufactured by MF appeared to induce lower antibody responses than those prepared using the TF method. Although the classical physicochemical characterization of liposomes through the DLS, ELS and TEM of blank liposomes showed comparable results of the two-particle species, the characterization of the two formulations through WAXS revealed some differences in the two bilayers’ chain-spacing distance and their hydration, resulting in a different orientation of the lipids. Diffraction peak fitting revealed that the lipid bilayers prepared using the MF procedure are well organized in a relatively homogeneously distributed and densely packed manner, with an average chain-spacing distance of 0.34 nm. However, the lipid bilayers produced via the rehydration of thin-film resulted in a larger average chain-spacing distance of 0.39 nm and were relatively more disordered. Additionally, the TF samples presented a significant water peak at q^0^ = 20.29 nm^−1^, suggesting a high hydration level of these lipid bilayers. It was reported that different hydration layers can directly influence the structure of the lipid bilayer [23]. These formulations result in statistically longer hydrocarbon inter-chain distances becoming conformationally more disordered.

A summary of the structural analysis of the liposomes is represented in Figure 13.

The difference in conformation of the lipid bilayer is likely to induce a difference in the interaction with the DNA vaccine at various N/P ratios. Previous studies have investigated the interaction of water molecules with the lipid bilayers, finding that their major site of residency is in phosphatidylcholine (PC) head groups and that the number of water molecules associated with head groups influences the packing of the acyl chain [24,25]. The full hydration of lipids can thicken a bilayer to up to 0.5 nm; therefore, the space between the lipids in the bilayer is highly influenced by its interaction with water molecules [26]. Hydration also could play a role. Given the hydrophilic nature of DNA, it might have a higher affinity for TF-manufactured liposomes due to their higher hydration of the bilayer, as how the composition and structure of liposomes influences the complexation of nucleic acids has already been shown [27]. We can already observe a difference in complexation with DNA when comparing TF- and MF-manufactured liposomes. In particular, the DLS results of MF showed a higher size and PDI for the 0.25:1 and 1:1 N/P ratios, suggesting a stronger presence of aggregates in the sample (Figure 4 and Figure 5). This can also be observed from the difference of the complexed samples in reaching the isoelectric point when complexed with DNA, suggesting a difference in the bilayer structure of the two liposome species [28]. The reason for this difference must be the manufacturing method. The thin-film rehydration method comprises a step of sonication, making it a high-energy technique, where the sample is subjected to harsh conditions to obtain small unilamellar vesicles (SUVs), while microfluidics manufacturing, on the other hand, consists of gentle and non-turbulent processing conditions. Based on the structural analysis, complexation behavior and in vivo immunological data, we hypothesize that sonication might influence the formation and structure of SUVs. When probe-sonication stress is applied, multi-layered vesicles (MLVs) formed after rehydration are disrupted and reconstituted. The bilayer fragments that are produced have a tendency to reassemble into SUVs, in spite of their bilayer being subjected to screw dislocation—a process known to modify the transport of molecules within the liposome and could have led to an increased entrapment of water in the bilayer [29,30]. The corpus of results within this study points out the critical importance of the multi-modal and multi-length scale characterization of physical and chemical properties of nanoparticles to predict their behavior in biological models [31].

## 5. Conclusions

In this paper, a successful liposome manufacturing method translation was carried out from the thin-film layer rehydration method to microfluidics. The systems were designed to have the same bulk physicochemical characteristics such as size, Zeta-potential, polydispersity and morphology. However, differences were observed after SARS-CoV-2 DNA vaccine complexation and in vivo mice studies. The more in-depth characterization of the structure through WAXS revealed differences in the state of hydration of the two bilayers, resulting in a different chain–chain distances between the lipids in the bilayer. In a pandemic setting, being able to manufacture accessible vaccines is vital. Transferring technologies among countries can highly impact the speed of distribution of the vaccine to the vast majority of the population [32]. When performing a tech transfer, minor or major changes in the manufacturing process can occur in order to adapt the method to the machines available [33]. This process of adaptation creates a new sample that needs to be structurally characterized to assess its similarity to the original one.

## Figures and Tables

**Figure 1 pharmaceutics-14-01427-f001:**
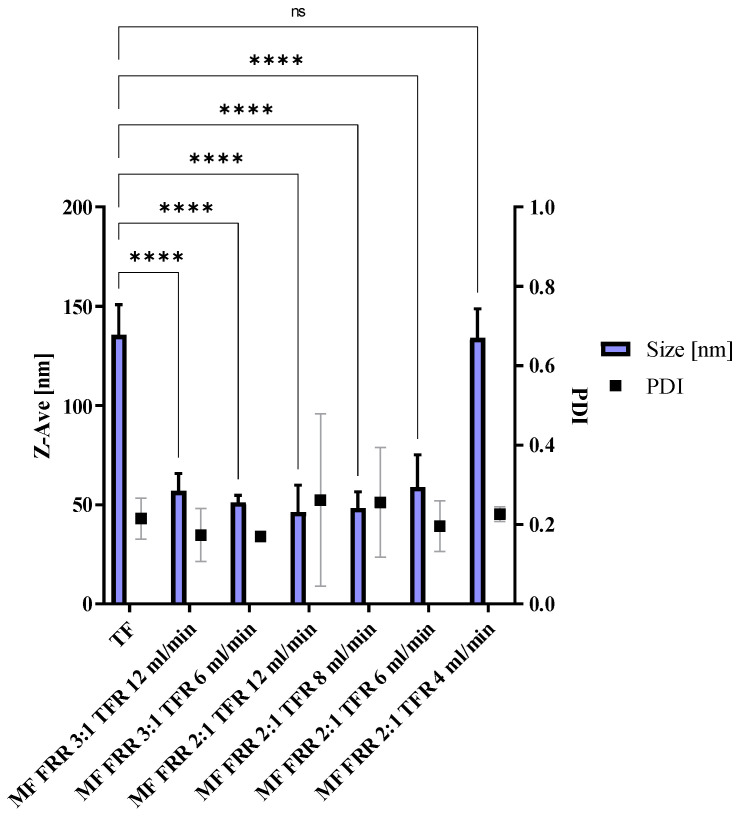
Size and polydispersity indices of liposomes prepared by a thin-film layer rehydration method and according to the microfluidics protocol, mean ± SD, *N* = 3. Unpaired Welch’s *t*-test. **** indicates *p* < 0.0001, ns shows non-significance. The test is meant to show differences among samples manufactured with microfluidics compared to the thin-film rehydration-manufactured particles control.

**Figure 2 pharmaceutics-14-01427-f002:**
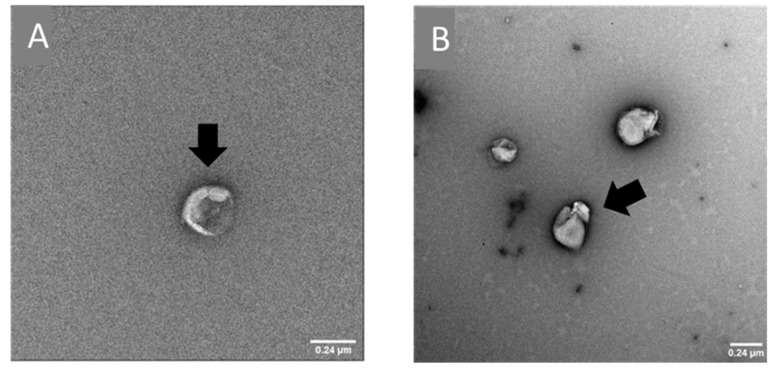
TEM images of blank DOTAP4 liposomes manufactured by thin-film layer rehydration (**A**), and by microfluidics (**B**). Bar = 0.24 µm.

**Figure 3 pharmaceutics-14-01427-f003:**
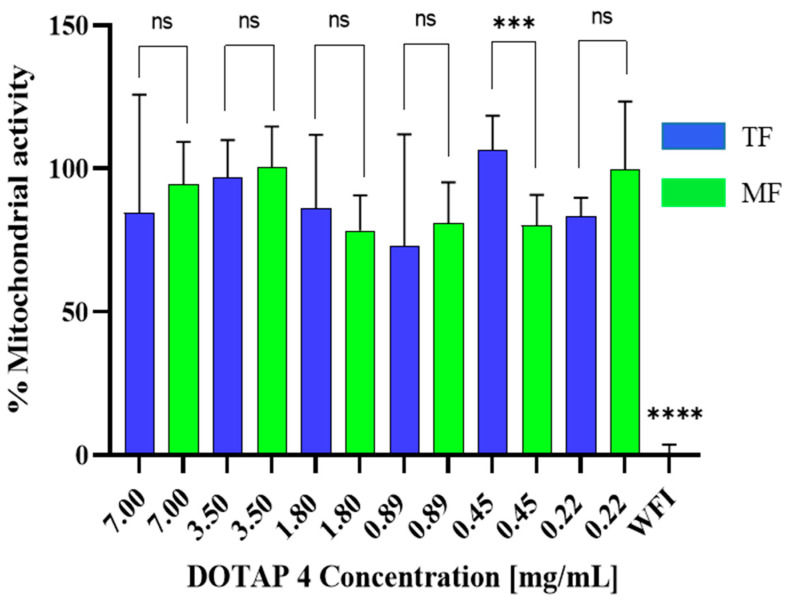
Toxicity of different concentrations of blank DOTAP 4 liposomes manufactured by microfluidics (MF) and thin-film (TF) rehydration method liposomes on RAW 264.7 cells, incubated on cells for 4 h. Data were normalized to non-treated cell results. WFI served as positive control. Data are represented as mean ± SD (*N* = 3). **** indicates *p* < 0.0001, *** *p* < 0.001, ns shows non-significance. All samples have a significant difference at *p* < 0.0001 with WFI control.

**Figure 4 pharmaceutics-14-01427-f004:**
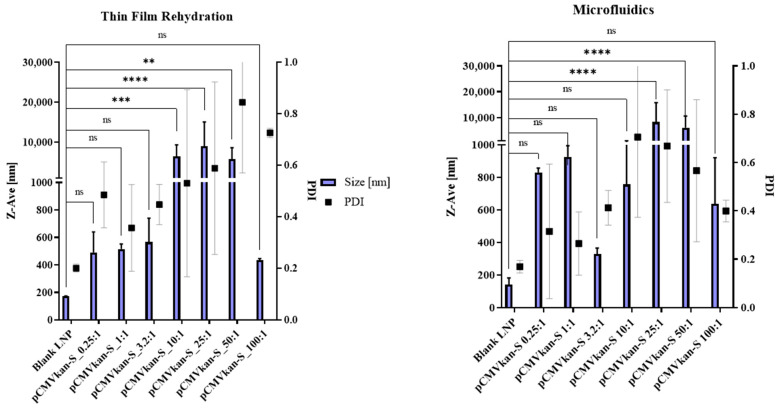
Size and size distribution of DOTAP4/pCMVkan-S complexes at different N/P ratios. Left graph shows the results obtained with thin-film layer rehydration-manufactured liposomes. Right graph shows the results obtained with microfluidics-manufactured liposomes. Data are represented as mean ± SD (*n* = 3). **** indicate *p* < 0.0001, *** *p* < 0.001, ** *p* < 0.01, ns shows non-significance when comparing sizes of liposomes complexed with DNA to blank particles.

**Figure 5 pharmaceutics-14-01427-f005:**
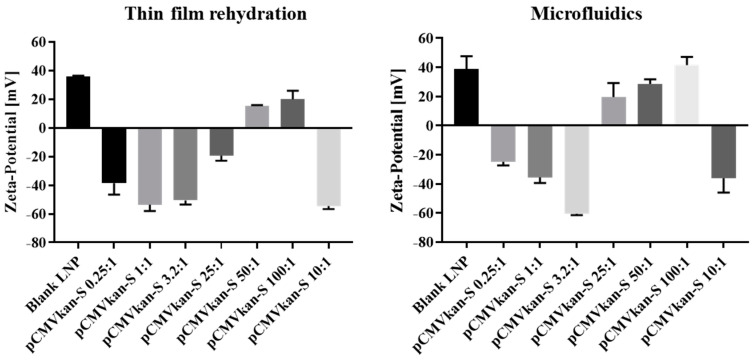
Zeta-potential results of DOTAP4/pCMVkan-S complexes at different N/P ratios. Graph on the left shows the results obtained with TF-manufactured liposomes. Graph on the right shows the results obtained with MF-manufactured liposomes. Data are represented as mean ± SD (*n* = 3).

**Figure 6 pharmaceutics-14-01427-f006:**
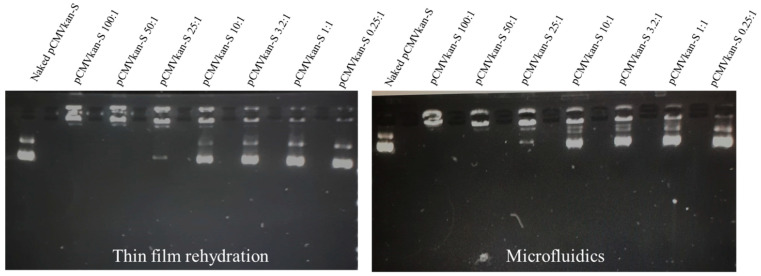
Agarose gel electrophoresis of naked pCMVkan-S in PBS or different N/P of DOTAP4/pCMVkan-S complex is shown. Image on the left and right show the results obtained from TF- and MF-manufactured liposomes, respectively.

**Figure 7 pharmaceutics-14-01427-f007:**
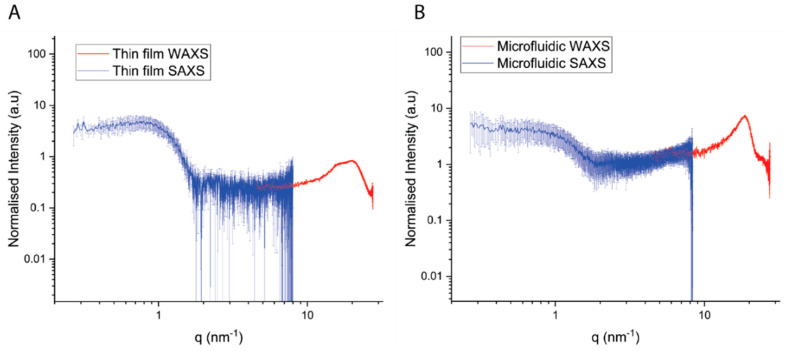
Normalized scattering intensity for lipid bilayer prepared via (**A**) TF, (**B**) MF. SAXS data are presented in blue, while WAXS data are in red.

**Figure 8 pharmaceutics-14-01427-f008:**
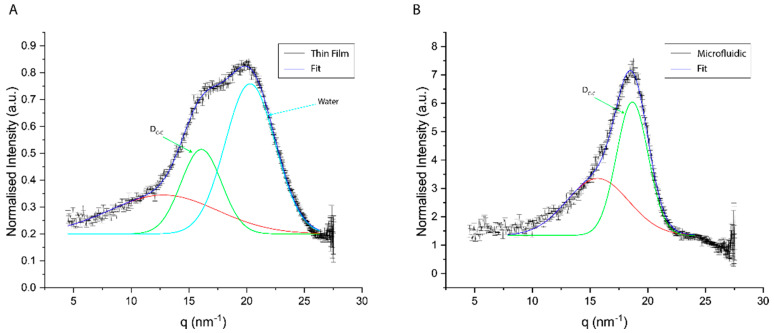
Fitting results of wide-angle scattering peak in both (**A**) MF and (**B**) TF samples. (D_C-C_)—Chain–Chain distance. Red line for background.

**Figure 9 pharmaceutics-14-01427-f009:**
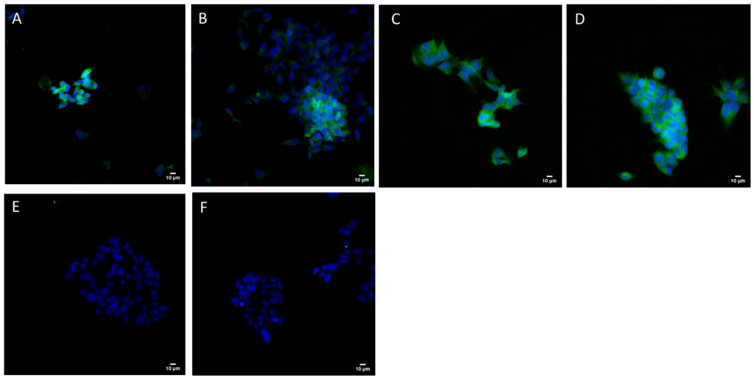
Intracellular SARS-CoV-2 S-protein expression. HEK293 were transfected with pCMVkan-S alone or as MF lipoplexes. (**A**) Lipofectamine/pCMVkan-S lipoplexes; (**B**) N/P 0.25:1 DOTAP4/pCMVkan-S; (**C**) N/P 1:1 DOTAP4/pCMVkan-S alone; (**D**) N/P 3.2:1 DOTAP4/pCMVkan-S; (**E**) naked pCMVkan-S; (**F**) only secondary Ab. Green = S protein, blue = nucleus. Images were obtained by confocal laser scanning microscopy, 40× magnification. Bar = 10 µm.

**Figure 10 pharmaceutics-14-01427-f010:**
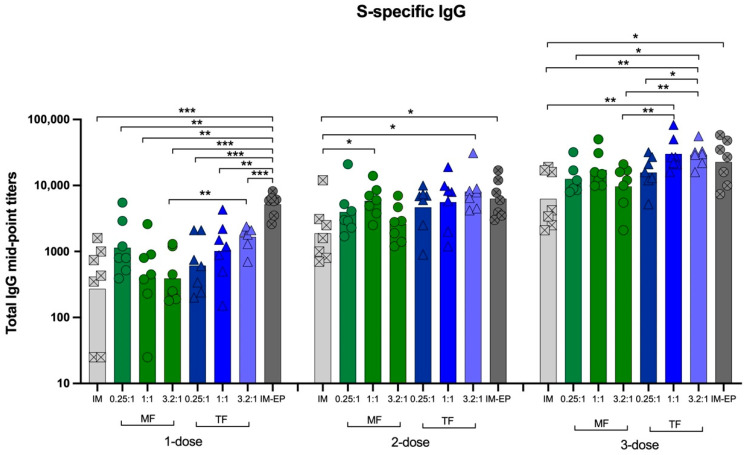
Midpoint titers of SARS-CoV-2 spike-specific total IgG analyzed at weeks 2, 4 and 6 in mice immunized intramuscularly with naked-pCMVkan-S (IM; crossed square), pCMVkan-S formulated with DOTAP4 at N/P ratios of 0.25:1, 1:1 and 3.2:1 manufactured by thin-film rehydration (TF; triangle) or microfluidics (MF; circle) methods and by using an electroporation device (IM-EP; crossed square). Each bar represents GMT of the midpoint IgG titers in each group (*n* = 7). *, **, and *** indicate *p* < 0.05, *p* < 0.01 and *p* < 0.001, respectively.

**Figure 11 pharmaceutics-14-01427-f011:**
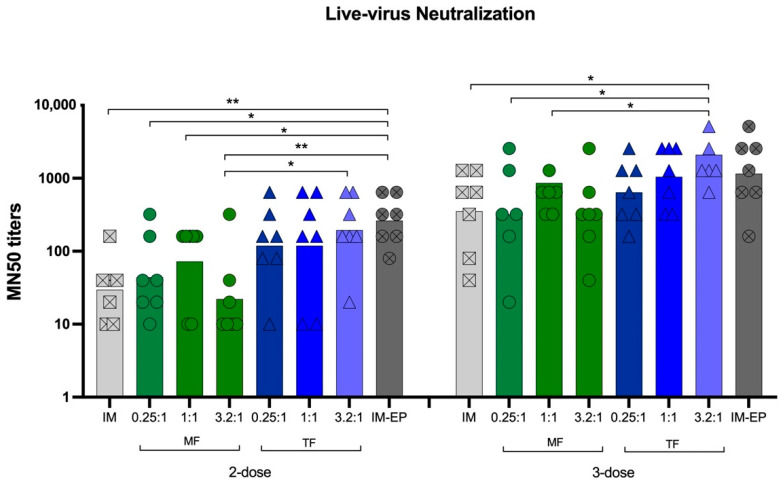
MN50 titer against live virus 2 weeks after receiving two- and three-dose vaccine, immunized intramuscularly with naked pCMVkan-S (IM; crossed square) and pCMVkan-S formulated with DOTAP4 at N/P ratios of 0.25:1, 1:1, and 3.2:1 manufactured using thin-film rehydration (TF; triangle) or microfluidics (MF; circle) methods and using an electroporation device (IM-EP; crossed circle). Each bar represents the GMT of the MN50 titers in each group (*n* = 7). * and ** indicate *p* < 0.05 and *p* < 0.01, respectively.

**Figure 12 pharmaceutics-14-01427-f012:**
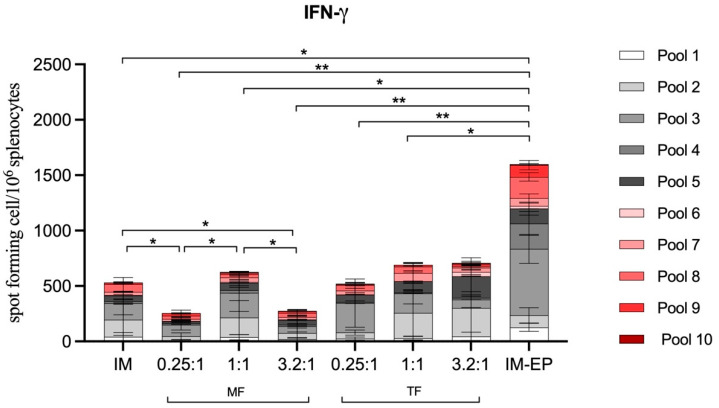
SARS-CoV-2 spike-specific T-cell responses analyzed by ELISpot. Mouse splenocytes were stimulated with pooled peptides from S1 (pool 1–5) and S2 (pool 6–10) regions. Each bar represents the sum of IFN-γ responses in mice (*n* = 7) immunized with different immunization strategies including naked pCMVkan-S (IM) and pCMVkan-S formulated with DOTAP4 at N/P ratios of 0.25:1, 1:1 and 3.2:1 manufactured using thin-film rehydration (TF) or microfluidics (MF) methods and using an electroporation device (IM-EP). * and ** indicate *p* < 0.05 and *p* < 0.01, respectively.

**Figure 13 pharmaceutics-14-01427-f013:**
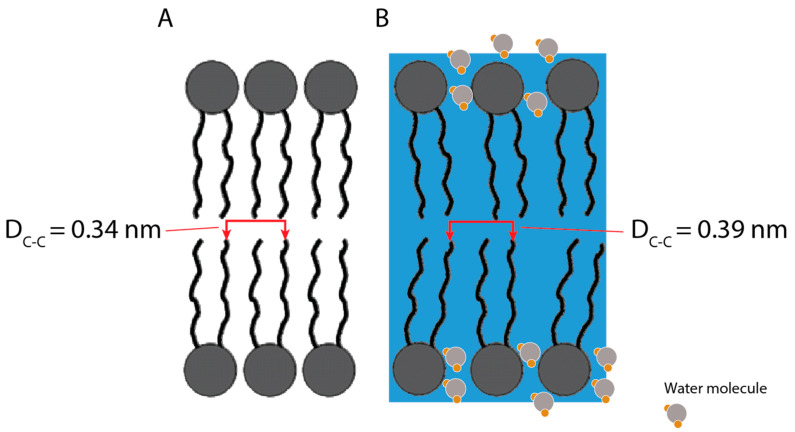
Schematic representation of the lipid structure in liposome mono-bilayer prepared via (**A**) microfluidics, (**B**) thin-film rehydration (blue square and stylized water molecules represent high hydration level). The inter-chain distance differences as well as the degree of disorder are exaggerated in the schematic to highlight the structural differences of both preparation methods.

**Table 1 pharmaceutics-14-01427-t001:** Lipoplex characteristics and summary of statistical analysis between same N/P ratios formulated with MF vs. TF particles. ^1^ The test performed is an unpaired Welch’s *t*-test. **** indicate *p* < 0.0001, *** *p* < 0.001, ** *p* < 0.01, ns shows non-significance.

Sample	Z-Ave (nm) ± SD TF	Z-Ave (nm) ± SD MF	*p*-Value Summary ^1^
Blank Liposomes	135.4 ± 15.3	134.0 ± 14.7	ns
pCMVkan-S 0.25:1	490.7 ± 149.7	829.8 ± 26.7	****
pCMVkan-S 1:1	515.8 ± 35.9	925.8 ± 68.5	****
pCMVkan-S 3.2:1	565.9 ± 174.4	329.9 ± 35.9	**
pCMVkan-S 10:1	6444 ± 2853	757.2 ± 252.3	***
pCMVkan-S 25:1	8974 ± 6056	8294 ± 7483	ns
pCMVkan-S 50:1	5776 ± 2788	6112 ± 4514	ns
pCMVkan-S 100:1	434.5 ± 11.9	636.8 ± 283.3	ns

**Table 2 pharmaceutics-14-01427-t002:** Fitting parameter of main scattering peaks in wide-angle. Chain–chain (*D*_C-C_) are expressed in nm^−1^ in reciprocal space and in nm in real space.

	Thin Film		Microfluidic
	Chain–Chain Spacing (*D*_C-C_)	Water Peak	Chain–Chain Spacing (*D*_C-C_)
*q*^0^ (nm^−1^)	16.06 ± 0.04	20.29 ± 0.03	18.66 ± 0.01
*D* (nm)	0.39 ± < 0.01	0.31 ± < 0.01	0.34 ± < 0.01
HWHM (nm^−1^)	1.57 ± 0.07	4.32 ± 0.05	1.37 ± 0.05

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
