# Peer review of "Translating a Thin-Film Rehydration Method to Microfluidics for the Preparation of a SARS-CoV-2 DNA Vaccine: When Manufacturing Method Matters"

_pharmaceutics, 2022, doi:10.3390/pharmaceutics14071427_

Round 1
Reviewer 1 Report
Peletta and coworkers present a manuscript comparing production methods of catatonic liposomes and their complexation with plasmid DNA carrying a gene encoding the SARS-CoV-2 spike protein. Overall, it appears that preparation of liposomes by the microfluids approach is comparable to the traditional thin film layer approach. A thorough review of the manuscript for English grammar is recommended.
Major Concerns:
1) Figure 7 – SAXS data presented in Figure 7A appear highly irregular and may be difficult to draw conclusions from.
Minor Concerns:
2) Lines 194-195 – indicate that the positive control using WFI causes cell lysis or similar description to improve clarity.
3) Line 211 – verify the incubation time of 4 hours is not supposed to be 24 hours. An incubation time of 4 hours seems short to produce significant quantities of detectible protein.
4) Figure 1 – was there any statistical analysis performed on the liposome preparations. Statistical differences were not indicated on the plot.
5) Line 227 – it is not clear what a “WST-1 assay” would measure. This type of assay is measuring mitochondrial activity as a marker for cell viability using the cell proliferation reagent WST-1.
6) Figure 3 – Figure is not very clear and does not demonstrate any differences in liposome preparation technique. A two-sample bar plot with samples at the same concentration being next to each other and each sample identified by a different color (or patterned bar for black and white figure reproduction). The statistical analysis is not clear. Unpaired T-tests should be performed between each preparation method at each concentration. Comparison with the positive control does not “show comparable results for the two formulations”.
7) Line 292 – “Size, distribution and Zeta-Potential…” it is not clear if the authors intend to state “Size distribution and Zeta-Potential…”
8) Figure 4 – Resolution of the figure should be increased for improved readability. Labels are barely readable without magnification and are of low resolution.
9) Line 317 – Figure presents size distribution data. It would make the legend more clear to state “Size distribution of DOTAP4/pCMVkan-S complexes…” instead of “DLS results…”
10) Lines 320-321 – The statistical analysis is not clear. Authors state that ANOVA was performed but do not indicate a post-hoc test to identify which samples were statistically different. Again, unpaired T-tests may be preferred between the two liposome preparation procedures at the same N/P ratios.
11) Figure 5 - Resolution of the figure should be increased for improved readability. Labels are barely readable without magnification and are of low resolution
12) Figure 6 – Presentation of the agarose gel is not standard and would be confusing to readers trying to discriminate between the supercoiled DNA normally on the bottom (here is the upper most band) and the open circular or linear forms normally on the top with decreased mobility (here is the lower most band). It is recommended to present the gel with the lanes at the top of the figure.
13) Figure 7 - Resolution of the figure should be increased for improved readability. Labels are barely readable without magnification and are of low resolution.
14) Figure 8 - Resolution of the figure should be increased for improved readability. Labels are barely readable without magnification and are of low resolution.
15) Line 356 – Text refers to Figure 2 but does not seem to reference the correct figure. Is this intended to be Figure 8?
16) Line 364 – Text refers to Figure 8 but should reference Figure 9.
17) Figure 9 – Images appear of low resolution and are blurry for confocal microscopy images.
18) Line 418 – Text refers to “S peptide pools”. Please describe these S peptide pools in the materials and methods with appropriate sequences or ranges provided in supplemental materials as needed.
19) Line 426 – A clear description of peptide pools is requested.
Author Response
We would like to thank you very much for your valuable comments. We hope these responses will answer your questions.
Comments and Suggestions for Authors
Peletta and coworkers present a manuscript comparing production methods of catatonic liposomes and their complexation with plasmid DNA carrying a gene encoding the SARS-CoV-2 spike protein. Overall, it appears that preparation of liposomes by the microfluids approach is comparable to the traditional thin film layer approach. A thorough review of the manuscript for English grammar is recommended.
Major Concerns:
- Figure 7 – SAXS data presented in Figure 7A appear highly irregular and may be difficult to draw conclusions from.
Response: Thank you for your comment. The irregularity is normal when displaying SAXS data, as the scattering plot is shown. You may find similar representation of data in the following articles:
Order Parameters and Areas in Fluid-Phase Oriented Lipid Membranes Using Wide Angle X-Ray Scattering. Mills et al, 2008 (1)
Size Determination of a Liposomal Drug by Small-Angle X‑ray Scattering Using Continuous Contrast Variation, Garcia-Diez et al. 2015 (2)
Using Liposomes as Carriers for Polyphenolic Compounds: The Case of Trans-Resveratrol, Bonechi et al. 2012 (3)
Minor Concerns:
- Lines 194-195 – indicate that the positive control using WFI causes cell lysis or similar description to improve clarity.
Response: The use of WFI to serve as the positive control in the cytotoxicity study is more clearly stated (Line 199-200).
- Line 211 – verify the incubation time of 4 hours is not supposed to be 24 hours. An incubation time of 4 hours seems short to produce significant quantities of detectible protein.
Response: Thank you for noticing, the typo was corrected to 24 h. (line 215)
- Figure 1 – was there any statistical analysis performed on the liposome preparations. Statistical differences were not indicated on the plot.
Response: Thank you for the comment, statistical analysis were added (line 284-286).
- Line 227 – it is not clear what a “WST-1 assay” would measure. This type of assay is measuring mitochondrial activity as a marker for cell viability using the cell proliferation reagent WST-1.
Response: The principle of WST-1 assay was included in the method (Line 194-195).
- Figure 3 – Figure is not very clear and does not demonstrate any differences in liposome preparation technique. A two-sample bar plot with samples at the same concentration being next to each other and each sample identified by a different color (or patterned bar for black and white figure reproduction). The statistical analysis is not clear. Unpaired T-tests should be performed between each preparation method at each concentration. Comparison with the positive control does not “show comparable results for the two formulations”.
Response: Figure 3 was revised with statistical analysis added to compare the two manufactured samples and with positive control. However, the main aim of this assay is to show the lack of important toxicity of the particles (mitochondrial activity >80%), rather than the similarity between the two. This is the reason why they were compared only to the positive control at first.
- Line 292 – “Size, distribution and Zeta-Potential…” it is not clear if the authors intend to state “Size distribution and Zeta-Potential…”
Response: Has been revised to “Size, size distribution and Zeta-Potential” (Line 308)
- Figure 4 – Resolution of the figure should be increased for improved readability. Labels are barely readable without magnification and are of low resolution.
Response: Figure resolution and labels were improved. The resolution is now 300 dpi, which complies with the journal's recommendation.
- Line 317 – Figure presents size distribution data. It would make the legend more clear to state “Size distribution of DOTAP4/pCMVkan-S complexes…” instead of “DLS results…”
Response: Revised in the legend of Figure 4 (line 355).
- Lines 320-321 – The statistical analysis is not clear. Authors state that ANOVA was performed but do not indicate a post-hoc test to identify which samples were statistically different. Again, unpaired T-tests may be preferred between the two liposome preparation procedures at the same N/P ratios.
Response: More information about the 2-way ANOVA test statistical analysis was added in the text. An additional unpaired Welch’s t-test was performed on the samples to compare their statistical differences. A table (Table 1) was added showing the results obtained. The multiple comparisons with the blank particles of the various samples is important as we need to take into consideration the size and state of aggregation of the particles for further studies rather than the actual differences between the two samples. However, we do think that adding this table is an added value and also helps the description of the figure in the text.
- Figure 5 - Resolution of the figure should be increased for improved readability. Labels are barely readable without magnification and are of low resolution
Response: Figure resolution was improved. The resolution is now 300 dpi, which complies with the journal's recommendation.
12) Figure 6 – Presentation of the agarose gel is not standard and would be confusing to readers trying to discriminate between the supercoiled DNA normally on the bottom (here is the upper most band) and the open circular or linear forms normally on the top with decreased mobility (here is the lower most band). It is recommended to present the gel with the lanes at the top of the figure.
Response: Agarose gel images were revised.
13) Figure 7 - Resolution of the figure should be increased for improved readability. Labels are barely readable without magnification and are of low resolution.
Response: Figure resolution was improved. The resolution is now 300 dpi, which complies with the journal's recommendation.
14) Figure 8 - Resolution of the figure should be increased for improved readability. Labels are barely readable without magnification and are of low resolution.
Response: Figure resolution was improved. The resolution is now 300 dpi, which complies with the journal's recommendation.
15) Line 356 – Text refers to Figure 2 but does not seem to reference the correct figure. Is this intended to be Figure 8?
Response: Revised to refer to Figure 8 (Line 426).
16) Line 364 – Text refers to Figure 8 but should reference Figure 9.
Response: Revised to refer to Figure 9 (Line 433).
17) Figure 9 – Images appear of low resolution and are blurry for confocal microscopy images.
Response: The images were exported from the confocal microscope and manipulated by Image J, which cannot be further adjusted.
18) Line 418 – Text refers to “S peptide pools”. Please describe these S peptide pools in the materials and methods with appropriate sequences or ranges provided in supplemental materials as needed.
Response: Spike peptide used in this study are the overlapping peptides spanning entire sequences of the SARS-CoV-2 spike protein. The information of spike peptide pools is now included in the Material and methods section (Line 246-246).
19) Line 426 – A clear description of peptide pools is requested.
Response: The information of spike peptide pools is now included in the Material and methods section (Line 246-246).
- Mills TT, Toombes GE, Tristram-Nagle S, Smilgies D-M, Feigenson GW, Nagle JF. Order parameters and areas in fluid-phase oriented lipid membranes using wide angle X-ray scattering. Biophysical journal. 2008;95(2):669-81.
- Garcia-Diez R, Gollwitzer C, Krumrey M, Varga Z. Size determination of a liposomal drug by small-angle X-ray scattering using continuous contrast variation. Langmuir. 2016;32(3):772-8.
- Bonechi C, Martini S, Ciani L, Lamponi S, Rebmann H, Rossi C, et al. Using liposomes as carriers for polyphenolic compounds: the case of trans-resveratrol. 2012.

Reviewer 2 Report
The authors report a liposome manufacture method translation from thin layer rehydration method to microfluidics, with both distinct SARS-COV-2 DNA vaccine complexation and in-vivo mice immunogenicity profiles, which is of interest for the readership of Pharmaceutics.
However, I recommend some amendments prior to acceptance:
1. Cationic lipids per se exert toxic effects on cells and - therefore - their use for nucleic acid complexation and gene delivery is a debatable approach. This dilemma and any progress made to minimise its impact should be indicated and briefly discussed, the more so it may contribute to distinct immunogenicity responses.
2. The authors mention as a major impetus for this study the search for an affordable pandemic vaccine, that is easy to produce, with a high manufacturing capacity and stable at temperature deviation. These properties are critical to make them available in low/middle income countries, particularly in tropical/subtropical regions. As a prime example, the far lower vaccination status concerning the mRNA Covid-19 vaccines of such countries compared with developed states is specified, which needs to be substantially increased.
However, I am not aware that those poor countries suffer a significant increase in morbidity and - particularly - mortality due to SARS-COV-2 infections. Accurate statistical data from respective national agencies and relevant papers need to be cited.
In contrast, some developed countries, and particularly such with a high vaccination status suffer a significant increase in morbidity/mortality after the vaccination campaign with the respective vaccines has started in 2021 (based on an emergency use approval) . This is clearly indicated in various governmental statistics, such as documented in the UK as "Deaths involving COVID-19 by vaccination status" by the Office for National Statistics (ONS, https://www.ons.gov.uk/) and worldwide in the WHO database (https://vigiaccess.org/).
Furthermore, the probability to get positively tested with COVID-19 (and thus recorded as infected patient) does not seem to be lower in vaccinated people than in unvaccinated ones. Refer, for example, to the current development in Portugal, where nearly all inhabitants are fully vaccinated.
In line with this notion, there are meanwhile more than thousand papers published in various peer-reviewed journals reporting a low vaccine effectiveness and/or adverse effects of the SARS-COV-2 vaccines as well as providing mechanistic insights into toxic effects of the vaccines and/or SARS-Cov-2 spike protein, e.g. Tetz and Tetz (2022) Prion-like Domains in Spike Protein of SARS-CoV-2 Differ across Its Variants and Enable Changes in Affinity to ACE. Microorganisms 2022 Jan 25; 10(2):280. doi: 10.3390/microorganisms10020280.
Last but not least, Pfizer has been forced by the US District Court Northern District of Texas (Az: Civil Action No. 4:21-cv-01058-P) to publish secret study details on its mRNA vaccine BNT162B2, which are accessible via www.phmpt.org.
Most relevant so far are study results summarised in a document (https://phmpt.org/wp-content/uploads/2021/11/5.3.6-postmarketing-experience.pdf) titled "CUMULATIVE ANALYSIS OF POST-AUTHORIZATION ADVERSE EVENT REPORTS OF PF-07302048 (BNT162B2) RECEIVED THROUGH 28-FEB-2021" stating that vaccine effectiveness is not proven as well as listing thousands of patients' cases having suffered from adverse effects (including death).
Taken together, it does not appear that a low vaccination status as in poor vs. developed countries represents a serious population health problem. Various kinds of evidence suggest rather a beneficial outcome for countries with a low vaccination status.
Accordingly, the respective reasoning in the manuscript should be adjusted.
Author Response
Reviewer 2
The authors report a liposome manufacture method translation from thin layer rehydration method to microfluidics, with both distinct SARS-COV-2 DNA vaccine complexation and in-vivo mice immunogenicity profiles, which is of interest for the readership of Pharmaceutics.
However, I recommend some amendments prior to acceptance:
- Cationic lipids per se exert toxic effects on cells and - therefore - their use for nucleic acid complexation and gene delivery is a debatable approach. This dilemma and any progress made to minimise its impact should be indicated and briefly discussed, the more so it may contribute to distinct immunogenicity responses.
- The authors mention as a major impetus for this study the search for an affordable pandemic vaccine, that is easy to produce, with a high manufacturing capacity and stable at temperature deviation. These properties are critical to make them available in low/middle income countries, particularly in tropical/subtropical regions. As a prime example, the far lower vaccination status concerning the mRNA Covid-19 vaccines of such countries compared with developed states is specified, which needs to be substantially increased.
However, I am not aware that those poor countries suffer a significant increase in morbidity and - particularly - mortality due to SARS-COV-2 infections. Accurate statistical data from respective national agencies and relevant papers need to be cited.
Search for paper that high morbid mortal associated with low vaccine supply
In contrast, some developed countries, and particularly such with a high vaccination status suffer a significant increase in morbidity/mortality after the vaccination campaign with the respective vaccines has started in 2021 (based on an emergency use approval) . This is clearly indicated in various governmental statistics, such as documented in the UK as "Deaths involving COVID-19 by vaccination status" by the Office for National Statistics (ONS, https://www.ons.gov.uk/) and worldwide in the WHO database (https://vigiaccess.org/).
Furthermore, the probability to get positively tested with COVID-19 (and thus recorded as infected patient) does not seem to be lower in vaccinated people than in unvaccinated ones. Refer, for example, to the current development in Portugal, where nearly all inhabitants are fully vaccinated.
In line with this notion, there are meanwhile more than thousand papers published in various peer-reviewed journals reporting a low vaccine effectiveness and/or adverse effects of the SARS-COV-2 vaccines as well as providing mechanistic insights into toxic effects of the vaccines and/or SARS-Cov-2 spike protein, e.g. Tetz and Tetz (2022) Prion-like Domains in Spike Protein of SARS-CoV-2 Differ across Its Variants and Enable Changes in Affinity to ACE. Microorganisms 2022 Jan 25; 10(2):280. doi: 10.3390/microorganisms10020280.
Last but not least, Pfizer has been forced by the US District Court Northern District of Texas (Az: Civil Action No. 4:21-cv-01058-P) to publish secret study details on its mRNA vaccine BNT162B2, which are accessible via www.phmpt.org.
Most relevant so far are study results summarised in a document (https://phmpt.org/wp-content/uploads/2021/11/5.3.6-postmarketing-experience.pdf) titled "CUMULATIVE ANALYSIS OF POST-AUTHORIZATION ADVERSE EVENT REPORTS OF PF-07302048 (BNT162B2) RECEIVED THROUGH 28-FEB-2021" stating that vaccine effectiveness is not proven as well as listing thousands of patients' cases having suffered from adverse effects (including death).
Taken together, it does not appear that a low vaccination status as in poor vs. developed countries represents a serious population health problem. Various kinds of evidence suggest rather a beneficial outcome for countries with a low vaccination status.
Accordingly, the respective reasoning in the manuscript should be adjusted.
Response:
Thank you very much for your invaluable comment. We do appreciate all those suggestions and also acknowledge the concern of using Covid-19 vaccine as they underwent an accelerated approval process to use in an emergency pandemic period. However, when the vaccines were massively introduced, more information were gathered and analyzed for their adverse events, efficacy, and effectiveness. Data from real-world studies that use the information from real pandemic situation regardless of experimental setting or assigned treatment are continously collected and reported. For example, a meta-analysis for SARS-CoV-2 vaccine effectiveness indicated that vaccines are highly protective against SARS-CoV-2-related diseases in real-world settings. Vaccines reduced hospitalization rate, admission to ICU and death at an effectiveness ranging from 65% to 98%, depending on the type of vaccine used (1). Another data study also showed that the number of new cases per million population, the reproduction rate of COVID-19, number of deaths per million population, and hospital and ICU patients per million population gradually decreased as the rate of vaccination coverage increased (over 60%) (2).
In addition, the Office for National Statistics (UK)analyzed the rate of death involving COVID-19 by vaccination status and reported that the death rate is increased in those who received the 2nd dose more than 6 months after the first. This indicates a possible waning protection by vaccination over time. However, the 3rd dose (booster) could lower the mortality rate, compared with unvaccinated people and those with just a first or second dose (3).
The shortage of vaccine supply that might happen in low- or middle-income countries could result in increasing both the spreading of the virus and the morbidity/mortality rate. For example, in Thailand, the vaccine could not be sufficiently supplied in mid 2021. By the end of September 2021, subjects who completed vaccination regimen was lower than 20% while approximately 70% of the population of Canada was vaccinated (4, 5). During the same period (September 2021), daily new deaths were approximately 0.36 and 0.1 per 100,000 individual in Thailand and Canada, respectively (6).
Taken together, these recent information could strengthen the notion of the need of vaccine supply and high vaccine coverage that might still be insufficient in some countries.
Last but not least, several studies on the risk-benefit assessment were consistently reporting that the benefits of vaccination against COVID-19 outweigh its risks (7, 8).

Reviewer 3 Report
This manuscript is assessing different liposome manufacturing methods which is a significant consideration for vaccine developers who need reproducible high-throughput manufacturing methods established as part of their commercialization strategy. Overall, manuscript is timely and well organized.
A few suggestions:
· All acronyms should be spelled out initially. Specific figures should also be referenced when discussed. For example, figures 11 and 12 are discussed but not referenced in the text.
· Minor grammar corrections are needed, particularly in the abstract.
· Figure can be combined to streamline the story. For example, figures 1 and 2 could be combined into a single figure. As could figures 4 and 5.
Author Response
Reviewer 3
This manuscript is assessing different liposome manufacturing methods which is a significant consideration for vaccine developers who need reproducible high-throughput manufacturing methods established as part of their commercialization strategy. Overall, manuscript is timely and well organized.
A few suggestions:
- All acronyms should be spelled out initially. Specific figures should also be referenced when discussed. For example, figures 11 and 12 are discussed but not referenced in the text.
Response: All acronyms are now spelled out upon first mention. Figures number 10, 11, and 12 are now referred to in the discussion .
- Minor grammar corrections are needed, particularly in the abstract.
Response: Grammatical errors and typos were corrected throughout the manuscript.
- Figure can be combined to streamline the story. For example, figures 1 and 2 could be combined into a single figure. As could figures 4 and 5.
Response: Thank you for your valuable comment. We acknowledge your suggestion to streamline the story and results. However, combination of figures might affect the resolution as suggested by reviewer 1. Thus, we decided to not combine the figures that might reduce readability.

Round 2
Reviewer 1 Report
Edits provided by the authors have improved the overall quality of the manuscript.
Author Response
Response: We have stated the concern of cytotoxicity of cationic liposomes (line 292-296). Mitochondrial activity was tested through a WST-1 assay to show cytotoxicity of the sample as seen in figure 3. In addition, microfluidics manufactured samples have also been tested after an incubation of 24 h with the cells at the same concentrations and showed no toxicity on RAW 264.7 cells (Fig.S1) (line 303-305).
SEE IMAGE IN PDF FILE
Fig S1. Toxicity of different concentrations of blank DOTAP 4 liposomes manufactured by microfluidics (MF) method liposomes on RAW 264.7 cells, incubated on cells for 24 h. Data were normalized to non-treated cell results.
Moreover, in our previous report, Peletta et al. 2021 (1) showed in the supplementary data (S5), the absence of toxicity of the thin film rehydration sample complexed with DNA at in vivo dose after incubation for 24 h. The image is reported below.
‘’Supplementary figure S5:
WST-1 assays on RAW 264.7 cells after 24h incubation with lipoplexes
SEE IMAGE IN PDF FILE
Figure S5: Mitochondrial activity of RAW 264.7 cells after 24 h incubation with lipoplexes at N/P ratios at 0.25:1 and 1:1 for both 100 µg and 1 µg of pCMVkan-S plasmid. Results are normalized to non-treated cells results. SDS 1% was used as a positive control.’’
We therefore suggest that at this stage of research sufficient toxicity data have been brought to the attention of the reviewers. We do understand that in case of further development a more in depth safety study will have to be carried out.
- Peletta A, Prompetchara E, Tharakhet K, Kaewpang P, Buranapraditkun S, Techawiwattanaboon T, et al. DNA Vaccine Administered by Cationic Lipoplexes or by In Vivo Electroporation Induces Comparable Antibody Responses against SARS-CoV-2 in Mice. Vaccines. 2021;9(8):874.
